# APC Loss Prevents Doxorubicin-Induced Cell Death by Increasing Drug Efflux and a Chemoresistant Cell Population in Breast Cancer

**DOI:** 10.3390/ijms24087621

**Published:** 2023-04-21

**Authors:** Casey D. Stefanski, Anne Arnason, Sara Maloney, Janna Kotsen, Elizabeth Powers, Jian-Ting Zhang, Jenifer R. Prosperi

**Affiliations:** 1Department of Biological Sciences, University of Notre Dame, Notre Dame, IN 46556, USA; cstefans@nd.edu (C.D.S.); aarnason8@gmail.com (A.A.); jkotsen@nd.edu (J.K.); epowers2@nd.edu (E.P.); 2Harper Cancer Research Institute, University of Notre Dame, Notre Dame, IN 46556, USA; saramaloney7196@gmail.com; 3Department of Biochemistry and Molecular Biology, Indiana University School of Medicine, South Bend, IN 46617, USA; 4Department of Cell and Cancer Biology, University of Toledo College of Medicine and Life Sciences, Toledo, OH 43606, USA; jianting.zhang@utoledo.edu

**Keywords:** Adenomatous Polyposis Coli, chemoresistance, doxorubicin, ABC transporters, STAT3, tumor initiating cells, breast cancer

## Abstract

Chemoresistance is a major health concern affecting cancer patients. Resistance is multifactorial, with one mechanism being the increased expression of ABC transporters (such as MDR1 and MRP1), which are drug efflux transporters capable of preventing intracellular accumulation of drugs and cell death. Our lab showed that the loss of *Adenomatous Polyposis Coli (APC)* caused an intrinsic resistance to doxorubicin (DOX), potentially through an enhanced tumor-initiating cell (TIC) population and the increased activation of STAT3 mediating the expression of MDR1 in the absence of WNT being activated. Here, in primary mouse mammary tumor cells, the loss of APC decreased the accumulation of DOX while increasing the protein levels of MDR1 and MRP1. We demonstrated decreased APC mRNA and protein levels in breast cancer patients compared with normal tissue. Using patient samples and a panel of human breast cancer cell lines, we found no significant trend between APC and either MDR1 or MRP1. Since the protein expression patterns did not show a correlation between the ABC transporters and the expression of APC, we evaluated the drug transporter activity. In mouse mammary tumor cells, the pharmacological inhibition or genetic silencing of MDR1 or MRP1, respectively, decreased the TIC population and increased DOX-induced apoptosis, supporting the use of ABC transporter inhibitors as therapeutic targets in APC-deficient tumors.

## 1. Introduction

Despite the medical advances in breast cancer, tens of thousands of American women die each year due to this illness. One of the issues that remain in providing adequate treatment is the establishment of chemotherapy resistance, which leads to treatment failure in over 90% of patients with metastatic cancer [1]. Chemotherapy is the standard of care for cancer when there is not an available targeted therapy option. Triple-negative breast cancer (TNBC) lacks the targetable receptors (estrogen, progesterone, and human epidermal growth factor 2 (HER2)) and therefore requires chemotherapy for treatment [2]. Given the prevalence of resistance, it is necessary to find a therapeutic target to help decrease the reliance on chemotherapy among these patients.

The tumor suppressor *Adenomatous Polyposis Coli* (*APC*) is altered (gene deletion, promoter methylation, or loss of protein) in up to 70% of sporadic breast cancers [3,4,5]. In addition, it was shown that patients with APC-deficient breast cancer lacked estrogen and progesterone receptors and demonstrated decreased overall survival as compared with patients with APC-positive breast cancer [4]. Using primary tumor cells from the *Apc^Min/+^* mouse model crossed to the polyoma middle T antigen (PyMT) transgenic model, our lab previously demonstrated decreased sensitivity to chemotherapeutics, including the commonly used therapy, doxorubicin (DOX) [6]. In addition, MMTV-PyMT;*Apc^Min/+^* cells did not exhibit increased Wnt activation demonstrating an Wnt-independent mechanism for resistance to DOX [7]. Understanding the cause of resistance to DOX is critical in developing combination therapies to thwart this resistance and improve patient outcomes.

Adenosine triphosphate (ATP)-binding cassette (ABC) transporters are well established drug efflux transporters that are elevated in cancers, preventing the accumulation of the drug within the cancer cells and, subsequently, apoptosis [8]. DOX can be effluxed by multiple ABC transporters, including multidrug resistance protein 1 (MDR1/ABCB1), breast cancer resistance protein (BCRP/ABCG2), or the multidrug resistance-associated protein 1 (MRP1/ABCC1) [9,10,11], making them an ideal therapeutic target. Unfortunately, previous clinical trials for these inhibitors were unsuccessful due to toxicity issues; however, these clinical trials did not utilize proper patient selection criteria to assess their efficacy, and a new generation of drugs has combated these earlier issues [12]. We have previously demonstrated that BCRP/ABCG2 protein levels are unchanged in MMTV-PyMT;*Apc^Min/+^* cells as compared with MMTV-PyMT;*Apc^+/+^* cells, even following DOX treatment [6]. We observed that the loss of APC increased the activation of signal transducer and activator of transcription 3 (STAT3) and the subsequent expression of MDR1 following DOX treatment [6,13]. While MRP1 has been shown to efflux DOX via co-transportation with glutathione, little is known about how the loss of APC affects MRP1 [14].

A small subgroup of cancer cells, called tumor-initiating cells (TICs), have been shown to contribute to recurrence and chemoresistance. Following conventional chemotherapy treatment, the surviving breast cancer cells are enriched in TICs and mesenchymal-like subpopulations [15]. One major reason for these cells being left behind is that TICs exhibit numerous chemoresistant phenotypic pathways, including quiescence, enhanced DNA damage repair, and increased expression of ABC transporters [16]. In fact, several ABC transporters are used as markers for TICs in certain cancer types (i.e., ABCG2 is a TIC marker in lung, liver, and prostate cancer) [17,18]. Therefore, targeting these ABC transporters might be advantageous for targeting this resistant subtype of cancer cells. Overall, these transporters have been shown to participate in DOX efflux and TIC survival, but whether it is APC’s effect on these transporters that is contributing to the observed DOX resistance remains unknown. In this study, we showed that MMTV-PyMT;*Apc^Min/+^* cells have increased MDR1 and MRP1 activity, resulting in decreased accumulation of DOX within the cells and decreased apoptosis. A combination therapy of DOX with the inhibitors of MDR1 and/or MRP1 resensitized the APC-deficient cells to DOX-induced apoptosis and decreased the chemoresistant TIC population.

## 2. Results

### 2.1. Decreased APC Expression in Breast Cancer

While APC was first established as a tumor suppressor in colon cancer, the loss of APC is seen in multiple different cancer types. To establish the clinical relevance of the loss of APC in breast cancer, by using the TCGA and CPTAC databases, we showed that APC mRNA and protein levels are decreased in primary human breast tumors compared with normal tissue (Figure 1A,B) [19,20]. As mentioned earlier, low APC expression is an indicator of poor prognosis and, based on our previous data [4,6], we hypothesized this is due to APC’s regulation of the chemotherapeutic response through the upregulation of ABC transporters. Increased levels of ABC transporters suggest increased cellular export of DOX. Using the innate fluorescence of DOX normalized to each genotype control, we tested whether there was a decrease in the intracellular DOX fluorescence in APC-deficient primary mouse mammary tumor cells. We found that MMTV-PyMT;*Apc^Min/+^* cells had decreased DOX accumulation compared with MMTV-PyMT;*Apc^+/+^* cells (Figure 1C). To account for proliferation, we evaluated the concentration of DOX after 1, 6, and 12 h of treatment. Minimal changes in the intracellular accumulation of DOX between the MMTV-PyMT;*Apc^+/+^* and MMTV-PyMT;*Apc^Min/+^* cells were observed at 1 and 6 h post-treatment. However, after 12 h of drug treatment, the MMTV-PyMT;*Apc^Min/+^* cells demonstrated a significant decrease in the intracellular accumulation of DOX compared with the control MMTV-PyMT;*Apc^+/+^* cells (Appendix A). This suggests that the observed decrease in DOX fluorescence is independent of the cells’ growth rates. Having previously demonstrated that the loss of APC increased the expression of MDR1 [6], we measured the expression of MRP1. We found increased MRP1 protein level in the MMTV-PyMT;*Apc^Min/+^* cells compared with the control treated MMTV-PyMT;*Apc^+/+^* cells (Figure 1D); however, DOX treatment did not further increase MRP1 protein levels in MMTV-PyMT;*Apc^Min/+^* cells.

### 2.2. Correlation between APC Expression Levels and ABC Transporters

To assess the correlation of APC expression levels to the prominent drug efflux transporters, MDR1 and MRP1, we used the patients’ tumor protein lysates and measured the protein levels of APC, MDR1, and MRP1 (Appendix A). In the patients’ tumor lysates, we found that decreased APC levels trended with an increase in MDR1 protein levels (r = −0.3176), but this was not significant (*p* = 0.4049) (Figure 2A). We found a nonsignificant positive trend between APC expression levels and MRP1 levels (0.4370; *p* = 0.2066) (Figure 2B). These data suggest that there may be some regulation of ABC transporters via the expression of APC; however, limited diverse molecular subtypes and the overall small sample of patient samples reduced the statistical power.

Using a panel of breast cancer cell lines, we measured the protein levels of APC, MDR1, and MRP1 to assess the correlation between APC and these transporters (Appendix A). Similar to what was seen in the patients’ tumor data, we found that there was no significant correlation (r = −0.3086; *p* = 0.2448) between APC deficiency and increased MDR1 expression (Figure 2C), but there was a negative trend. There was a positive correlation (0.4950; *p* = 0.05124) between increased APC and increased MRP1 protein levels (Figure 2D), which supported the positive trend that we saw in the patients’ samples. In addition, using the human cell line panel, we explored the correlation between APC expression levels and the IC50 of DOX after 24 h of treatment (Appendix A). We found that low APC expression levels trended (r = −0.2371) with a higher DOX IC50 at 24 h, but not significantly (*p* = 0.3948) (Appendix A). These data reflect the trends that we observed among the patients’ samples and demonstrated that the loss of APC could affect the expression of these drug efflux transporters. It was surprising to see that increased MRP1 expression levels correlated with increased APC expression levels, since we found the opposite in the mouse cell lines (Figure 1D); however, the trend in MDR1 expression levels was similar across the patients’ data, the human cell lines, and the mouse cells, suggesting that this may be the better target.

### 2.3. MDR1 and MRP1 Activity Caused Decreased Intracellular DOX Accumulation

MDR1 and MRP1 are known DOX efflux transporters that prevent the intracellular accumulation of DOX and thus DOX-induced apoptosis. Therefore, it is the activity of the ABC transporter, as opposed to its expression, that is essential in promoting resistance. To determine whether increased MDR1 and MRP1 activity contributed to decreased DOX accumulation, we used a combination treatment of DOX with chemical or genetic inhibition of MDR1 and MRP1. Using DOX with the MDR1 inhibitor Valspodar increased the mean DOX fluorescence in MMTV-PyMT;*Apc^Min/+^* cells compared with DOX treatment alone (Figure 3A). As a MRP1-specific inhibitor is not currently available, we transfected both MMTV-PyMT;*Apc^+/+^* cells and MMTV-PyMT;*Apc^Min/+^* cells with either siNeg or siMRP1, and their knockdown was confirmed by Western blotting (Appendix A). The silencing of MRP1 in combination with DOX treatment resulted in an increase in DOX fluorescence in MMTV-PyMT;*Apc^Min/+^* cells compared with mock and siNeg-transfected MMTV-PyMT;*Apc^Min/+^* cells (Figure 3B). While this disagreed with the human correlation data (Figure 2), this may be due to the difference in the species and mechanisms for the loss of APC. Use of the MRP1 and MDR1 inhibitor Reversan in combination with DOX also increased the accumulation of DOX in the MMTV-PyMT;*Apc^Min/+^* cells to similar levels compared with the treatment with DOX alone (Figure 3C). It should be noted that the Reversan treatment also influenced the accumulation of DOX in the MMTV-PyMT;*Apc^+/+^* cells (Figure 3C). Our data suggested that the activity of both MDR1 and MRP1 was responsible for the reduced intracellular DOX fluorescence in MMTV-PyMT;*Apc^Min/+^* cells.

### 2.4. Inhibition of MDR1 and MRP1 Alone and in Combination Decreased the Population of TIC as Measured by ALDH^high^ Cell Populations

Our lab had previously shown that the loss of APC increased TICs as measured by an increase in the ALDH^high^ cell population [6]. ALDH’s activity is an established murine marker for tumor-initiating cells in breast cancer. As ABC transporters are upregulated in TICs, contributing to their chemoresistant phenotype [21], we questioned whether disrupting MDR1 and/or MRP1 would affect this cell population. In an Aldefluor assay to measure the activity of ALDH, ALDH^High^ cells converted the Aldefluor substrate into its fluorescent product. The change in the shift was determined between the control and the treated population within each genotype. Therefore, ALDH^High^ cells had an increased shift in fluorescence compared with cells treated with the ALDH inhibitor DEAB. This percent shift from the control treated cells to the percent shift after MDR1 inhibition (Figure 4A), siMRP1 treatment (Figure 4B), or Reversan treatment (Figure 4C) decreased, which demonstrates a decrease in the ALDH^High^ cell population compared to untreated MMTV-PyMT;*Apc^Min/+^* cells. While perturbing MDR1 and MRP1 decreased the ALDH^High^ population in the APC-deficient cells, it did not affect the MMTV-PyMT;*Apc^+/+^* cells, nor did it reduce the ALDH^High^ population to that of MMTV-PyMT;*Apc^+/+^* cells. This suggests that while the activity of MDR1/MRP1 contributed to the ALDH^High^ population’s survival, other signaling modalities also influenced the ALDH^High^ and, presumably, the TIC population.

### 2.5. Increased ABC Transporters Cause DOX Resistance

The increased activity of ABC transporters is a common mechanism of chemoresistance. To ascertain whether this activity contributed to the observed DOX resistance in MMTV-PyMT;*Apc^Min/+^* cells, the cells were treated with DOX and the MDR1 inhibitor Valspodar, and the expression levels of the apoptotic marker cleaved caspase 3 (CC3) were measured (Figure 5A). We found that the combination treatment increased DOX-induced apoptosis in MMTV-PyMT;*Apc^Min/+^* cells compared with treatment with DOX alone. The level of apoptosis with the combination treatment was similar to the level seen for DOX-only and co-treatment in MMTV-PyMT;*Apc^+/+^* cells (Figure 5A).

In addition to MDR1, DOX can also be effluxed out of the cell through the ABC transporter, MRP1. Cells were transfected and then treated with DOX 24 h prior to collection to measure the amount of CC3. We found that MMTV-PyMT;*Apc^Min/+^* cells transfected with siMRP1 had increased DOX-induced CC3 expression levels as compared with mock- or siNeg-transfected MMTV-PyMT;*Apc^Min/+^* cells. The levels observed in the combination treatment were similar to the amount of apoptosis observed in DOX-treated and mock-transfected MMTV-PyMT;*Apc^+/+^* cells (Figure 5B).

Next, we tested whether dual inhibition using the MRP1 and MDR1 inhibitor Reversan would restore DOX sensitivity. We treated cells for 24 h with DOX or Reversan alone or the combination of the two. While the combination treatment increased DOX-induced apoptosis in MMTV-PyMT;*Apc^Min/+^* cells compared with the single treatments in MMTV-PyMT;*Apc^Min/+^* cells, the expression levels of CC3 were similar to those under the DOX-only treatment in the MMTV-PyMT;*Apc^+/+^* cells but not under the combination treatment (Figure 5C). In addition, the combination treatment increased the expression of CC3 in the MMTV-PyMT;*Apc^+/+^* cells compared with the single-drug treatments (Figure 5C), similar to what was observed in the drug accumulation studies (Figure 3C). These data demonstrate that the activity of both MDR1 and MRP1 contributes to the DOX resistance seen in MMTV-PyMT;*Apc^Min/+^* cells.

## 3. Discussion

Overall, we found that the loss of APC affects the activity of the ABC transporter, promoting cancer cells’ survival following DOX treatment. We found, in human cell lines and patients’ tumor lysates, that APC deficiency trended with higher MDR1 but lower MRP1 levels. The murine cell line that we used in our studies is heterozygous for APC, containing a copy of both full-length and truncated APC, with no observed loss of heterozygosity [7]. Truncated APC has been shown to have gain-of-function roles [22], which could contribute to the opposing results we found in our mouse model compared with patients, who rarely have truncating mutations. Additionally, our small sample size reduced the statistical power of the study and prevented us from separating the correlation data of the expression levels into subtypes; however, previously, MDR1 and MRP1 have been shown to be elevated in breast cancer, especially in TNBC, which relies on chemotherapy as the first line of treatment [23]. The observed negative trend between APC and MDR1 makes this a better targeted therapy option in APC-deficient cancer. This is best seen in the data from the mouse cell line, where we observed increased activity of MDR1 and MRP1, but the inhibition of MDR1 affected DOX-induced apoptosis in an APC-dependent manner. This could be further validated with the use of a human cell line model of APC loss to demonstrate these pathways causing DOX resistance. While the inhibition of MDR1 in combination with DOX increased the mean DOX fluorescence in MMTV-PyMT;*Apc^Min/+^* cells to similar levels to those observed in DOX-treated MMTV-PyMT;*Apc^+/+^* cells, the same combination treatment in MMTV-PyMT;*Apc^+/+^* cells resembled the single-drug treatment. This demonstrates that the loss of APC was specific in driving this enhanced MDR1 efflux activity, whereas manipulating the activity of MRP1 did not solely affect APC-deficient cells. The increase in DOX fluorescence following the inhibition of MDR1 and MRP1 silencing was similar to that in the MMTV-PyMT;Apc^Min/+^ cells. However, there was a significant increase in DOX fluorescence in MMTV-PyMT;Apc^+/+^ cells only upon MRP1 silencing in combination with DOX. This suggests that the expression of APC was not the only factor regulating the activity of MRP1, and it was not the main driver of APC-deficient DOX resistance, ultimately supporting the inhibition of MDR1 as a potential therapeutic target in APC-deficient breast cancer.

While previous clinical trials for ABC transporter inhibitors were inconclusive, indirect inhibition of these ABC transporters may be a safer and effective therapeutic alternative. Previously, our lab showed that STAT3 was responsible for the DOX-induced MDR1 expression levels seen in MMTV-PyMT;*Apc^Min/+^* cells. While we also know that STAT3 is increased in primary tumors compared with normal tissues (Appendix A) and that STAT3 increased the expression and activity of MDR1 (Appendix A), how the loss of APC causes upregulated activation of STAT3 is not understood. Understanding how the ABC transporters are upregulated in cancer is imperative for new therapeutic approaches. ABC transporter inhibitors have shown a mild response in the clinic, with toxicity and drug–drug interactions being an issue; therefore, being able to find new approaches to target these drug efflux transporters is paramount [24]. One such approach may include the regulation of microRNAs, which have been shown to regulate chemoresistance [25]. For instance, in breast cancer, decreased miR-200c was correlated with patients not responding to neoadjuvant chemotherapy. This study also showed that targeting miR-200c influenced the expression of MDR1 and intracellular accumulation of DOX [26,27]. In addition, miR-124-3p, targeting *ABCB1*, was shown to induce apoptosis in MCF7/Dox cells [28]. Another avenue for targeting ABC transporters and drug resistance is through the long noncoding RNA (lncRNA). The lncRNA MALAT1 has been shown to regulate the expression of MRP1 and MDR1 by targeting STAT3, leading to cisplatin resistance [29]. This demonstrated that these noncoding RNAs are one example that could contribute to DOX resistance in APC-deficient cells and could be used as therapeutic targets.

Targeting the ABC transporters showed advantages outside of affecting the normal drug-resistant efflux function but also decreasing the ALDH^High^ population, which was indicative of the chemoresistant TICs. Another avenue that may be therapeutically beneficial would be to see how the loss of APC supports the TIC population and potential therapeutic targets to prevent chemoresistance. TICs have numerous multidrug resistance (MDR) pathways to survive chemotherapy [30]; therefore, one method might not be enough to abolish this population, as demonstrated by the Aldefluor assay, where the inhibition of ABC transporters or STAT3 did not decrease the ALDH^High^ population (Figure 4 and Appendix A) to the levels of MMTV-PyMT;*Apc^+/+^* cells. While targeting this drug-resistant population is promising, very few clinical trials specifically target TICs and, therefore, most clinical studies are not measuring the effect of these treatments on the TIC population. However, a preclinical trial showed that treatment with an ALDH inhibitor increased the sensitivity to chemotherapy and radiotherapy in ALDH^High^/CD44^+^ breast cancer cells [31]. Understanding the direct role of the loss of APC in regulating TICs and how to target them, will be essential in eliminating this highly chemoresistant population to prevent tumor regression.

Targeting either the upstream regulators of ABC transporters or TIC populations is a new avenue of therapeutic care that might prove instrumental in improving patient outcomes. Utilizing these combination therapies in an additional human cell line and in vivo mouse models will be important next steps in demonstrating their potential clinical use.

## 4. Materials and Methods

### 4.1. Database Analysis

The expression of APC in primary tumors compared with normal tissues was analyzed using the University of Alabama at Birmingham’s Cancer Data Analysis Portal (UALCAN) data portal [19,20]. The Cancer Genome Atlas (TCGA) samples were from the breast invasive carcinoma dataset consisting of 114 normal tissues and 1097 primary tumor samples. The Clinical Proteomic Tumor Analysis Consortium (CPTAC) samples were from the breast cancer subset consisting of 18 normal tissue and 125 primary tumor samples.

### 4.2. Cell Culture

MMTV-PyMT;*Apc^+/+^* and MMTV-PyMT;*Apc^Min/+^* cells were isolated as previously described [7] and grown in RPMI 1640 media with 10% fetal bovine serum (FBS), 1% penicillin/streptomycin (P/S), and 1:5000 plasmocin (Invitrogen, Waltham, MA, USA). HCC1954, HCC1937, HCC38, DU4475, HCC1187, HCC1569, HCC1395, HS578T, and ZR-75-1 cells were also maintained in RPMI 1640 media with 10% FBS, 1% P/S, and 1:5000 plasmocin. MCF-7, MDA-MB-231, and MDA-MB-468 cells were maintained in DMEM media supplemented with 10% FBS, 1% P/S, and 1:5000 plasmocin. BT20 and BT474 cells were maintained in DMEM/F-12 media with 10% FBS, 1% P/S, and 1:5000 plasmocin. SKBR3 cells were maintained in RPMI 1640 media supplemented with 10% FBS, 1% P/S, 1 mM sodium pyruvate, and 2 mM L-glutamine. All cells listed above were obtained from the American Type Culture Collection (ATCC) (Manassas, VA, USA). KTB21-hTERT, KTB37-hTERT, and KTB34-hTERT cells (acquired from the Komen Tissue Bank) were maintained in DMEM/F-12 media supplemented with 5% FBS, 1% P/S, 0.4 μg/mL hydrocortisone, 5.0 μg/mL insulin, 20 ng/mL EGF, and 24 mg/L adenine [32]. KTB cells were passaged only 10 times. All cells were routinely passaged and maintained at 37 °C with 5% CO_2_.

### 4.3. Patient Samples

Tissue samples from primary mammary tumors were acquired from the Harper Cancer Research Biorepository (IRB# 20-06-6095). The expression levels of ER, PR, and HER2 were recorded and are provided in Table 1. Samples were from patients aged 40–83, with no information about their prior treatment.

### 4.4. Drug Treatment

Murine cells were plated and treated 24 h later with each chemotherapeutic agent or a solvent control: doxorubicin (500 nM, MP Biomedicals, LLC), Valspodar (10 μM, Sigma, St. Louis, MO, USA), Reversan (5 μM, Sigma), or A69 (10 μM [33,34]). Drug treatments were performed for 24 h unless otherwise indicated.

### 4.5. Cell Viability

Cell viability was measured using the CellTiter-Blue^®^ Cell Viability kit (Promega, Madison, WI, USA). Human cells were seeded on a 96-well plate and were treated with a range of doxorubicin concentrations. After 24 h of treatment, CellTiter-Blue^®^ was added to each well and incubated for 4 h following the manufacturer’s protocol. Plates were read on the SpectraMax M3 plate reader.

### 4.6. SiRNA Transfection

Murine cells were seeded on a 6-well plate and transfected 24 h later with 4 μg of siNeg (Sigma, SIC001) or siMRP1 (Sigma, EMU053861) using 7.5 μL of Lipofectamine 2000, prepared in Opti-MEM (Gibco, Waltham, MA, USA). The transfection reagent was removed 24 h later, fresh media were added, and the cells were incubated for an additional 48 h prior to flow cytometry or protein isolation. Mock-treated cells only received Lipofectamine 2000.

### 4.7. Western Blotting

*Mouse Lysates* Whole-cell lysates were isolated using a lysis buffer (20 mM Tris-HCl, 150 mM NaCl, 1% Triton-X, 0.5% NP-40, 50 mM NaF, 1 mM Na_3_VO_4_, 5 mM sodium pyrophosphate, 0.2 mM PMSF, 1X protease inhibitor cocktail (Fisher, Waltham, MA, USA), and 1X phosphatase inhibitor cocktail 2 (Sigma, St. Louis, MO, USA)), and a BCA assay (Thermo, Waltham, MA, USA) was performed to determine the protein concentrations. Samples (protein and sample buffer) were boiled for 5 min prior to gel loading (except for when probing for MRP1 and MDR1, as per the manufacturer’s recommendations). Prepared samples of 5–30 μg of protein were separated by SDS-PAGE gel and transferred onto an Immobilon P membrane (Millipore, St. Louis, MO, USA). The membranes were blocked in 5% NFDM for 1 h at room temperature. The blots were probed with the following primary antibodies: MRP1 (1:1000, Abcam ab260039, Boston, MA, USA), MDR1 (1:1000, Thermo MA5-32282, Waltham, MA, USA), cleaved caspase 3 (1:1000, CST 9661, Danvers, MA, USA), and actin (1:25,000, Sigma A1978-200, St. Louis, MO, USA). Blots were developed using either the Clarity or Max Clarity reagent on a ChemiDoc MP Imaging System (Bio-Rad, Hercules, CA, USA) and analyzed using ImageJ software (NIH) to measure the densitometry.

*Human Lysates* Whole-cell lysates were isolated using a lysis buffer (50 mM Tris (pH 7.5), 150 mM NaCl, 0.5% NP-40, 1.0 mM EDTA, 0.2 mM PMSF, 1Xprotein inhibitor cocktail (Fisher, Waltham, MA, USA)). Each tumor was placed in the lysis buffer and the tissue was dissociated using a Dounce homogenizer. The lysates were centrifuged and the supernatant was collected without the lipid layer. Protein concentrations were measured using a BCA assay (Thermo, Waltham, MA, USA), and 10–30 μg of the proteins were separated by a BioRad Mini-PROTEAN TGX Precast 4–20% gradient gel and transferred overnight at 30 V onto an Immobilon P membrane (Millipore, St. Louis, MO, USA). Samples (protein and sample buffer) were boiled for 5 min prior to gel loading (except for when probing for MRP1, as per the manufacturer’s recommendations) prior to gel loading. Following blocking with 5% nonfat dry milk (NFDM) in 1X TBST for 1 h, the blots were probed with the following primary antibodies: MRP1 (1:1000, Abcam ab260038, Boston, MA, USA), MDR1 (1:1000, CST 13978S, Danvers, MA, USA), APC (1:500, Novus NB100-91662, Littleton, CO, USA), vinculin (1:1000, CST 13901S, Danvers, MA, USA), and actin (1:25,000, Sigma A1978-200, St. Louis, MO, USA). The blots were developed using either the Clarity or Max Clarity reagent on a ChemiDoc MP Imaging System (Bio-Rad. Hercules, CA, USA) and analyzed using ImageJ software (NIH) to measure the densitometry.

### 4.8. Aldefluor Assay

The enzyme activity of aldehyde dehydrogenase (ALDH) was determined using the Aldefluor™ kit (Stem Cell Technologies, Vancouver, BC, Canada) according to the manufacturer’s protocol. Samples were prepared as previously described [6].

### 4.9. Measurement of Intracellular DOX 

The intracellular accumulation of DOX was measured using a protocol for the accumulation of mitoxantrone as previously described [35], but with a modification of 488 nm for excitation and >670 nm for emission for detecting doxorubicin. Briefly, the cells were seeded for 24 h prior to treatment. The cells were washed with 1X PBS, centrifuged, and washed twice more and resuspended in PBS. Flow cytometry (Cytotomics FC 500, Beckman Coulter, Brea, CA, USA) was used to measure fluorescence with the FL2 channel. The mean fluorescence of DOX was analyzed using FlowJo flow cytometry data analysis software (Tree Star, Ashland, OR, USA).

### 4.10. Statistical Analysis

All values are reported as the mean ± SD. Two-tailed T-tests were performed on the TCGA and CPTAC samples. Two-tailed Pearson correlations were used to analyze the relationships among the IC50s of APC, MDR1, MRP1, and DOX. Data from multiple treatment groups were compared using one-way ANOVA with a post hoc Tukey’s t-test. All graphs and statistics were performed using GraphPad Prism version 9.3.1. for Windows (La Jolla, CA, USA).

## Figures and Tables

**Figure 1 ijms-24-07621-f001:**
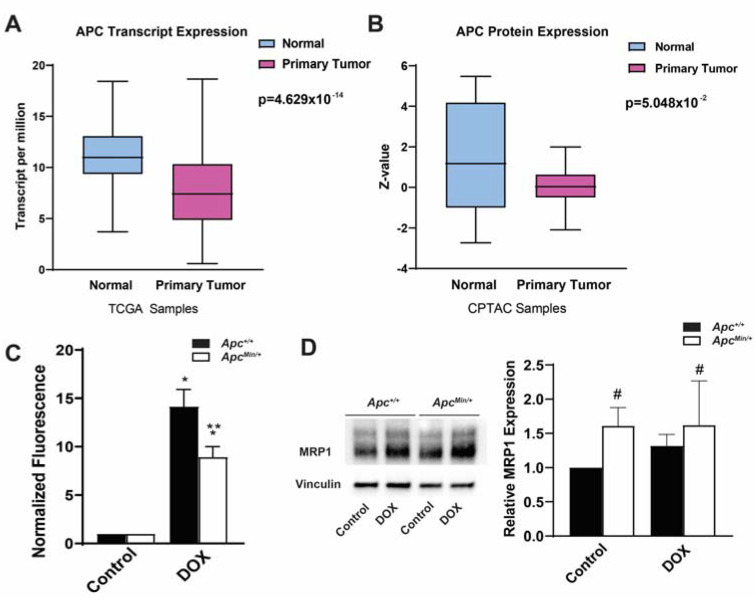
APC expression levels and correlations to ABC transporters in breast cancer patients. (**A**) TCGA samples showed APC transcription in primary tumors compared with normal breast tissue, with APC transcript levels lower in primary tumors. (**B**) CPTAC samples showing the APC protein levels in primary tumors and normal breast tissue, with APC expression levels being lower in primary tumors. (**C**) Intracellular DOX fluorescence, as a measure of the intracellular accumulation of the drug, demonstrated that MMTV-PyMT;*Apc^Min/+^* cells have decreased intracellular accumulation of DOX, suggesting increased drug efflux. Data were normalized to the genotype control. (**D**) Representative Western blots and quantification of Western blots demonstrating increased MRP1 protein levels in APC-deficient cells, with the DOX treatment having no effect. * *p* < 0.05, comparing treated cells with control-treated cells; ** *p* < 0.05, comparing MMTV-PyMT;*Apc^Min/+^* cells with MMTV-PyMT;*Apc^+/+^* cells within the same treatment; # *p* < 0.05, comparing the MMTV-PyMT;*Apc^+/+^* cells and MMTV-PyMT;*Apc^Min/+^* cells with or without treatment with the control-treated MMTV-PyMT;*Apc^+/+^* cells.

**Figure 2 ijms-24-07621-f002:**
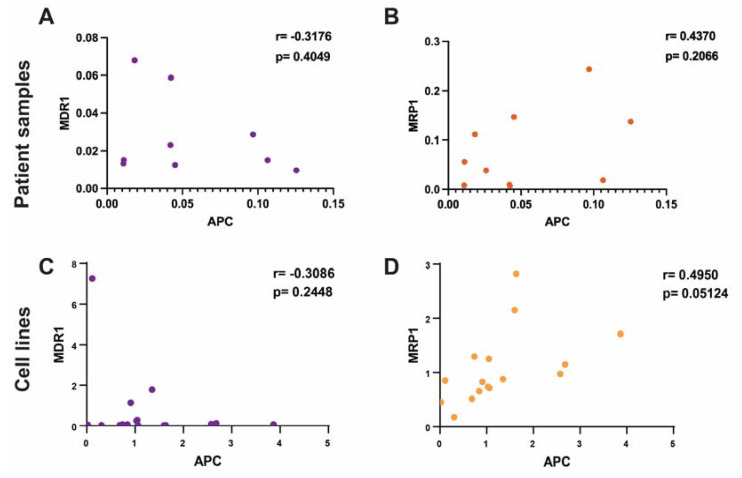
Correlation of APC with DOX sensitivity and the expression levels of MDR1 and MRP1 in human breast cancer. (**A**) Using patient tumor samples, the expression of APC was correlated with MDR1 expression levels. Protein levels were normalized to the normal epithelial cell lines. Lower levels of APC protein trended with higher levels of MDR1 protein (Pearson’s r = −0.3176) but this trend was not significant (*p* = 0.4049). (**B**) In the patients’ tumor samples, the protein level of APC was not significantly correlated with MRP1 protein levels. Higher APC protein levels trended with higher MRP1 protein levels, but this was not significant (Pearson’s r = 0.4370; *p* = 0.2066). (**C**) In a panel of human breast cancer cell lines, the expression levels of APC were correlated with the protein expression levels of MDR1 and MRP1. Protein levels were normalized to the average of the normal breast epithelial cell lines. Decreased APC levels trended with increased MDR1 levels (Pearson’s r = −0.3086), but the trend was not significant (*p* = 0.2448). (**D**) Increased APC protein levels correlated with increased MRP1 protein levels (Pearson’s r = 0.4950; *p* = 0.05124).

**Figure 3 ijms-24-07621-f003:**
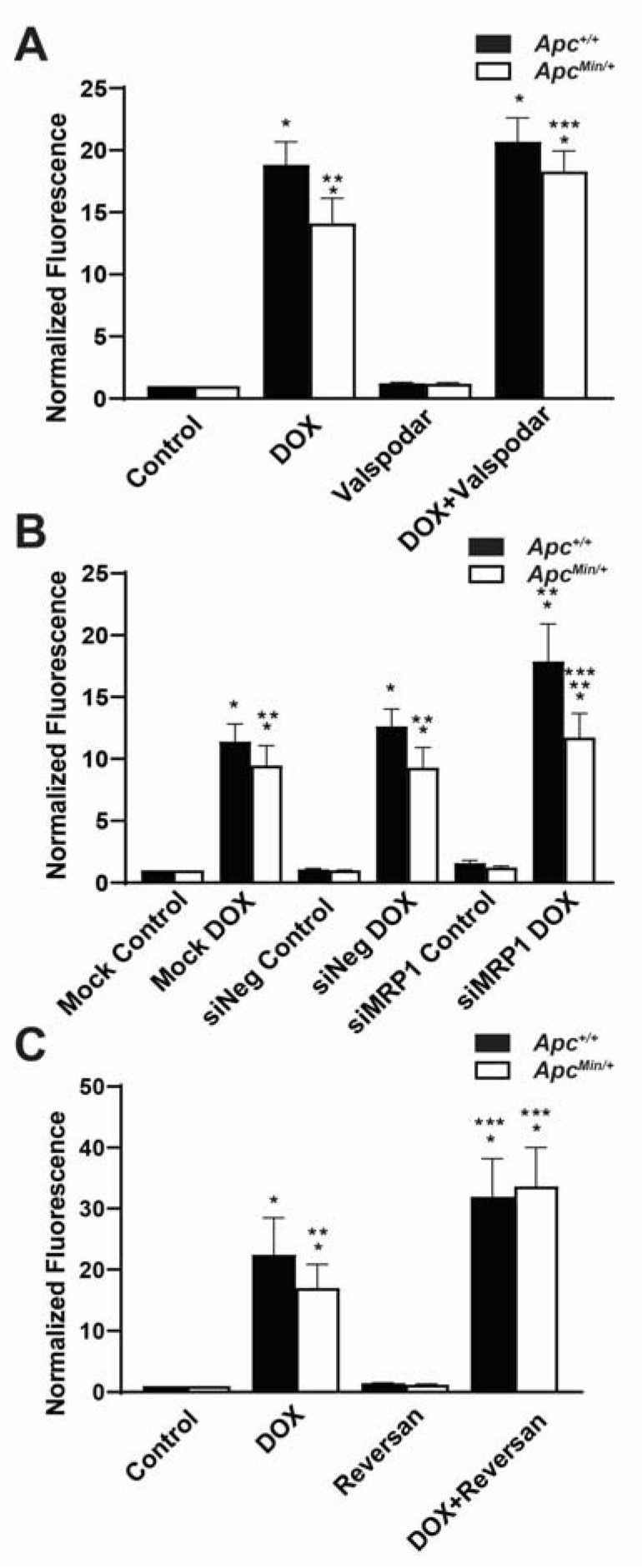
Increased MDR1 and MRP1 activity effluxes DOX extracellularly in APC-deficient cells. (**A**) MDR1 inhibition restores the accumulation of DOX in MMTV-PyMT;*Apc^Min/+^* cells comparable to the accumulation of DOX in MMTV-PyMT;*Apc^+/+^* cells. The mean fluorescence intensity (MFI) was normalized to that of the genotype control. (**B**) The accumulation of DOX increased in MMTV-PyMT;*Apc^Min/+^* and MMTV-PyMT;*Apc^+/+^* cells following MRP1 silencing. MFI was normalized to the genotype control. (**C**) Dual MDR1 and MRP1 inhibition increased intracellular DOX fluorescence in MMTV-PyMT;*Apc^Min/+^* cells and MMTV-PyMT;*Apc^+/+^* cells. MFI was normalized to the genotype control. * *p* < 0.05, comparing treated cells with control-treated cells; ** *p* < 0.05, comparing MMTV-PyMT;*Apc^Min/+^* cells with MMTV-PyMT;*Apc^+/+^* cells within the same treatment/transfection; *** *p* < 0.05, comparing the combination treatment with the DOX-only treatment.

**Figure 4 ijms-24-07621-f004:**
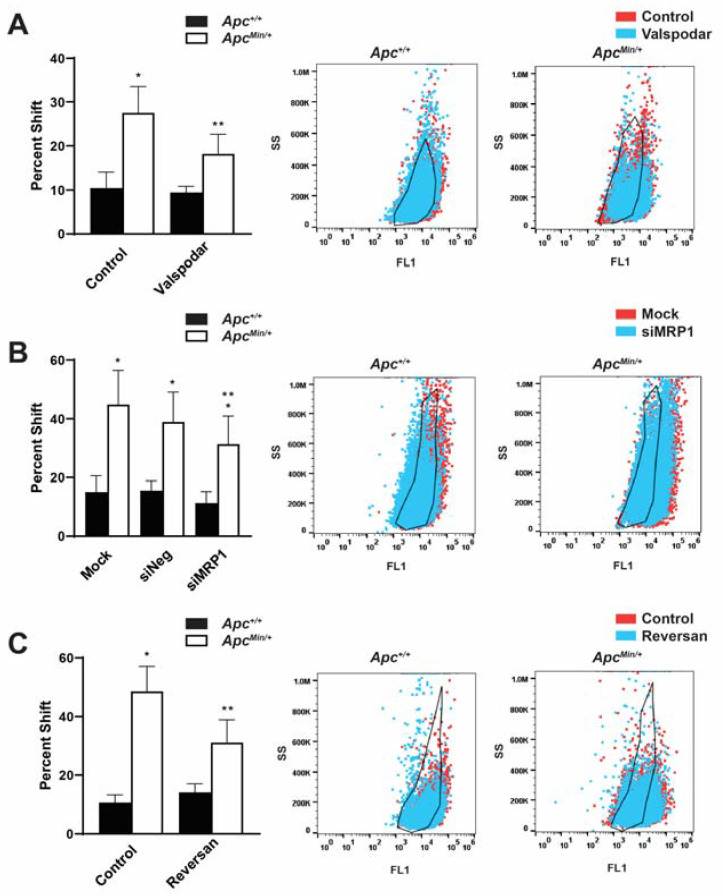
Inhibition MDR1 and MRP1 enhances the ALDH^High^ TIC population. (**A**) The ALDH^High^ cell population in MMTV-PyMT;*Apc^Min/+^* cells decreased following the inhibition of MDR1. Representative population shifts demonstrated the change between the control and the Valspodar treatment compared with the black outline of the untreated control within each genotype. (**B**) MRP1 silencing decreased the ALDH^High^ population compared with mock-transfected MMTV-PyMT;*Apc^Min/+^* cells. The representative population shifts demonstrated the change between mock and siMRP1 transfection compared with the black outline of the untreated control within each genotype. (**C**) Dual inhibition of MDR1 and MRP1 decreased the MMTV-PyMT;*Apc^Min/+^* cells’ ALDH^High^ population. The representative population shifts demonstrated the change between the control and Reversan treatment compared with the black outline of the untreated control within each genotype. This demonstrates a dependence on these transporters in the ALDH^High^ population, which is enhanced in APC-deficient cells. * *p* < 0.05, comparing MMTV-PyMT;*Apc^Min/+^* cells with MMTV-PyMT;*Apc^+/+^* control-treated/mock-transfected cells; ** *p* < 0.05, comparing treated or siMRP1-transfected MMTV-PyMT;*Apc^Min/+^* cells with control-treated or mock-transfected MMTV-PyMT;*Apc^Min/+^* cells.

**Figure 5 ijms-24-07621-f005:**
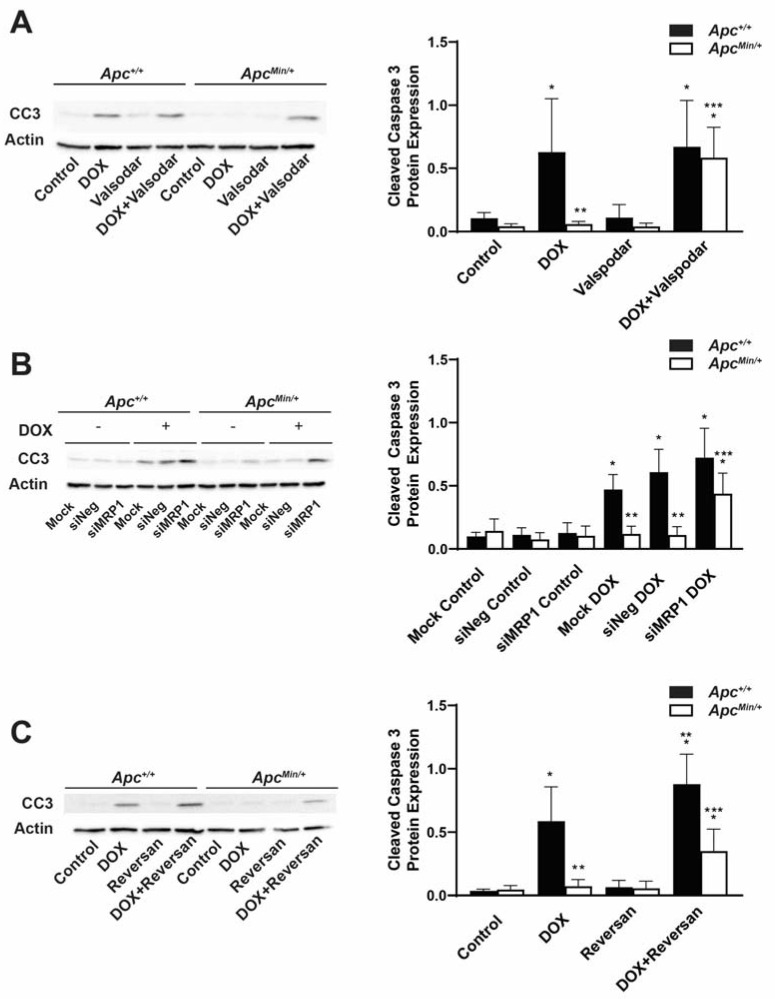
Inhibition of the ABC transporter restores DOX-induced apoptosis in MMTV-PyMT;*Apc^Min/+^* cells. (**A**) DOX-treated MMTV-PyMT;*Apc^Min/+^* cells have less CC3 compared with DOX-treated MMTV-PyMT;*Apc^+/+^* cells. MDR1 inhibition sensitized MMTV-PyMT;*Apc^Min/+^* cells to DOX-induced apoptosis. Densitometry was performed, and CC3 was normalized to Actin. (**B**) Quantification and representative Western blots showed that silencing MRP1 increased DOX-induced CC3 in MMTV-PyMT;*Apc^Min/+^* cells as compared with the DOX-only treatment, similar to the CC3 levels in DOX-treated MMTV-PyMT;*Apc^+/+^* cells. Densitometry was performed, and CC3 was normalized to Actin. (**C**) Treatment with a MDR1/MRP1 inhibitor plus DOX increased the CC3 in MMTV-PyMT;*Apc^Min/+^* cells compared with DOX-only treatment. The combination treatment also increased DOX-induced apoptosis in MMTV-PyMT;*Apc^+/+^* cells compared with the DOX-only treatment. Densitometry was performed, and CC3 was normalized to Actin. * *p* < 0.05, comparing treated cells to control-treated cells; ** *p* < 0.05 as compared with DOX-treated MMTV-PyMT;*Apc^+/+^* cells; *** *p* < 0.05, comparing combination treatment in MMTV-PyMT;*Apc^Min/+^* cells and MMTV-PyMT;*Apc^+/+^* cells.

**Table 1 ijms-24-07621-t001:** Characteristics of the patients’ tumors.

Sample	ER	PR	HER2
ND978	+	−	−
ND1063	+	+	−
ND1114	+	+	−
ND1116	+	+	+
ND1167	+	+	−
ND1195	+	+	−
ND1205	+	+	−
ND1210	−	−	−
ND1222	+	+	−
ND1230	+	+	−
ND1240	+	+	−

## Data Availability

The data presented in this study are available in this article and its associated Appendix A.

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
