# Peer review of "APC Loss Prevents Doxorubicin-Induced Cell Death by Increasing Drug Efflux and a Chemoresistant Cell Population in Breast Cancer"

_ijms, 2023, doi:10.3390/ijms24087621_

Round 1
Reviewer 1 Report
This manuscript investigates the role of MDR1 and MRP1 activity in mediating drug efflux and chemoresistance with loss of APC. While the topic is important and the findings are interesting, the study should be improved with additional samples to demonstrate reproducibility and a clarification of TICs.
Specific Comments:
- The source of the patient tumor lysates is not described. It is also unclear if the samples were taken from patients who received chemotherapy or treatment naive.
- There is an extensive panel of tumor cell lines that were queried for expression and IC50 response to doxyrubicin, but none were used in the functional in vitro studies. It would be pertinent to include some examples of low and/or high APC expressing human cell lines to verify the phenotypic changes observed in the mouse line used throughout the paper. Otherwise most of the study is really only reflective of a single cell line and therefore lacks rigor. This is especially critical given the heterogeneity of breast cancer.
- The data presented in Figure 2 should be labeled with "cell line" or "patient" lysates.
- In Figure 4, it seems more appropriate to describe ALDH activity increases rather than describing it as changes in the TIC population. TIC phenotypes are not assessed in this study, only ALDH activity, which can vary and does not necessarily indicate tumor initiation capacity.
- the combination treatment in Figure 5C does not seem as effective as the single inhibition experiments in A and B. Can the authors discuss explanations for this?
- Given the lack of focus on TICs in this paper, the third paragraph of the discussion seems somewhat irrelevant.
- Two major weaknesses of the study are the lack of additional cell lines to show reproducibility and no in vivo models to show translational relevance.
Reviewer 2 Report
In this study, the authors showed that APC loss was associated with an increase in MDR1 and MRP1 activity, resulting in decreased DOX accumulation within the cells and decreased apoptosis. Combination therapy of DOX with inhibitors of MDR1 and/or MRP1 re-sensitized the APC-deficient cells to DOX-induced apoptosis and reduced the chemoresistant TIC population.Given the prevalence of DOX resistance in solid tumors chemotherapy, the topic of the manuscript is interesting and up to date; the manuscript is written clearly, and the presentation of results follows a coherent line; various methods and techniques are used. However, I have some comments on the current version of the manuscript to share with the authors.
- Since this study is focused on breast cancer, I suggest including "breast cancer" in the title.
- Figures 3 and 4 should be enlarged because they were too small to see the relevant features.
- The authors must clarify the relationship between APC loss and MRP1 expression levels, as the manuscript contains contradictory results (see lines 154 and 170)
- I recommend adding the molecular weight of the protein marker ladder to the full-length uncropped blots. I propose that they be included in the Supplementary Material.
Author Response
We thank this reviewer for the constructive comments on our manuscript and have addressed their questions point by point.
- Since this study is focused on breast cancer, I suggest including "breast cancer" in the title.
Thank you for the suggestion. We have made this change.
- Figures 3 and 4 should be enlarged because they were too small to see the relevant features.
Thank you for the suggestion. We have made these changes.
- The authors must clarify the relationship between APC loss and MRP1 expression levels, as the manuscript contains contradictory results (see lines 154 and 170)
We apologize for the confusion on the results from Figure 2D. We have now clarified that there is a positive relationship between APC and MRP1 in the human samples and human cell lines (Figure 2B and D), while we see the opposite in the mouse cell lines (Figure 1D). We’ve also emphasized, in the text, that this suggests potentially a more important role for MDR1. Changes can be seen from lines 141-158.
- I recommend adding the molecular weight of the protein marker ladder to the full-length uncropped blots. I propose that they be included in the Supplementary Material.
This has been done. We have not included them in supplemental material, as the journal requires them to be submitted as a separate PDF file.
Round 2
Reviewer 1 Report
If additional cell lines are not going to be included in the manuscript to validate the functional studies, this should be discussed as a limitation of the study.
Author Response
Thank you for this comment. We have added that additional cell lines would be important for validating the results of our study in lines 294-295 and 346-347. in the discussion